# Impact of nutrient warning labels on Colombian consumers' selection and identification of food and drinks high in sugar, sodium, and saturated fat: A randomized controlled trial

**Mercedes Mora-Plazas[1‡], Isabella Carolyn Aida Higgins[2,3‡], Luis Fernando Gomez[4], Marissa G. Hall[3,5], Maria Fernanda Parra[4], Maxime Bercholz[2], Nandita Murukutla[6], Lindsey Smith Taillie[2,7] ***

1 Departamento de Nutrición Humana, Universidad Nacional de Colombia, Bogotá, Colombia, United States of America, 2 Carolina Population Center, University of North Carolina, Chapel Hill, North Carolina, United States of America, 3 Department of Health Behavior, Gillings School of Global Public Health, University of North Carolina, Chapel Hill, North Carolina, United States of America, 4 Facultad de Medicina, Pontificia Universidad Javeriana, Bogotá, Colombia, United States of America, 5 Lineberger Comprehensive Cancer Center, University of North Carolina, Chapel Hill, North Carolina, United States of America, 6 Vital Strategies, New York, New York, United States of America, 7 Department of Nutrition, Gillings School of Global Public Health, University of North Carolina, Chapel Hill, North Carolina, United States of America

‡ MMP and ICAH are contributed equally to this work as co-first authors.
* taillie@unc.edu

## Abstract

### Objective

This study assessed the impact of nutrient warnings on product selection and ability to identify food products high in nutrients of concern in Colombia.

### Methods

In an online experiment (May-June 2023), Colombian adults were randomized to a nutrient warning, guideline daily amounts (GDA), Nutri-Score, or no-label condition (n = 8,004). Participants completed selection tasks between two fruit drinks labeled according to their condition, one high in sugar and one not. Next, participants answered questions about products high in sugar, sodium, and/or saturated fat ("high-in" product). Finally, they selected which label would most discourage them from consuming a high-in product.

### Results

Fewer participants (17%) exposed to the nutrient warning indicated they would purchase the high-sugar fruit drink compared to Nutri-Score (27%, Holm-adjusted (adj) p<0.001) and no label conditions (31%, adj p<0.001); there were no differences between the nutrient warning and GDA label (14%, adj p = 0.087). Compared to the nutrient warning, the GDA label was slightly more effective at helping consumers identify which drink was high in sugar

**Data Availability Statement:** All data and do files are available from Open Source Framework (https://osf.io/5vbzm).

**Funding:** Funding support was provided by Bloomberg Philanthropies. The funder did not play any role in the study design, data collection and analysis, decision to publish, or preparation of the manuscript.

**Competing interests:** The authors have declared that no competing interests exist.

(89% versus 92%, adj p<0.001), while the Nutri-Score and no-label conditions were less effective. Compared to all other conditions, nutrient warnings were more effective at helping participants identify that products were high in nutrients of concern, were more effective at decreasing intentions to purchase these high-in products and were perceived as more effective. Nutrient warnings were most often selected as the label that most discouraged consumption.

## Conclusions

Nutrient warnings are a promising policy to help consumers identify and discourage consumption of products high in nutrients of concern.

## Trial registration

*Trial Registration*: NCT05783726.

## Background

Obesity and diet-related non-communicable diseases have become great health challenges, posing risks to the health and lives of individuals, the well-being of families, and economic development [1, 2]. Colombia is not immune to such health challenges; according to the Colombian National Nutritional Health Surveys (ENSIN) conducted in 2010 and 2015, the prevalence of overweight and obesity increased 5.6 percentage points in school-aged children (5–12 years old), 2.4 percentage points in adolescents (13–17 years old), and 5.2 percentage points in adults (18–64 years old) [3, 4]. Furthermore, a recent study assessing the diseases for which obesity and overweight are risk factors found that in Colombia, obesity and overweight contribute to approximately 9.4% of disability-adjusted life years, 17% of years lived with a disability, and 3% of years of life lost [5].

There is compelling evidence about the link between the shift from consumption of unprocessed foods to ultra-processed foods and the increase in obesity and diet-related non-communicable diseases [6–9]. Ultra-processed foods are generally low in beneficial nutrients like fiber, protein, micronutrients, and bioactive compounds [10–12] and tend to be high in nutrients related to chronic diseases such as sugar, sodium, and saturated fat ("high-in" products) [13]. In Colombia, from 2009 to 2014, per capita sales of ultra-processed foods and beverages increased by 7.7% [14] and an analysis of responses to the ENSIN 2005 and ENSIN 2015 found that both children and adults had worsening diets with increased consumption of "high in" products including sugar-sweetened beverages and processed meats [15].

To address the rising prevalence in overweight, obesity and diet-related non-communicable diseases, scholars, advocates and policymakers are increasingly calling for policies to communicate the health risks of consuming "high in" products and to discourage their consumption [16, 17]. Front-of-package labels have emerged as one promising policy to guide and influence consumers to make healthier food choices and purchasing decisions [18, 19]. Many countries in the world have applied different voluntary or mandatory front-of-package labels such as guideline daily amounts (GDA) labels, Nutri-Score labels, and nutrient warning labels (hereinafter referred to as nutrient warnings). Currently, in Latin America, nutrient warnings are the most common labeling system and are required in Peru, Uruguay, Chile, Mexico, Argentina, Brazil, and Colombia [20]. Emerging evidence suggests that of the different front-of-package

labels, nutrient warnings may be most effective at helping consumers to identify "high in" products and discourage them from selecting such products [21, 22].

Colombia is one of the most recent countries to adopt mandatory front-of-package nutrient warnings. On July 30, 2021, the president signed a nutrient warning bill into law [23], and on December 13, 2022, a resolution was passed to implement the law [24]. The law requires nutrient warnings on packages for excess sodium, sugar, saturated fat, trans fat, and if they contain artificial sweeteners. The law is currently being implemented in Colombia, with a final enforcement date of June 2024. Before this law, there was public debate between advocacy coalitions, the Colombian government, and the food industry with regards to which type of front-of-package label should be implemented, with the food and health coalition advocating for a nutrient warning, and the food industry advocating for alternative front-of-package labels, like the GDA label [25, 26].

Thus, data on which type of front-of-package label is most impactful at shifting consumers' selections, perceptions, and intentions to purchase is timely and critical. While a recent online randomized controlled trial in Colombia, which assessed perceptions of and reactions to different nutrient warning designs, concluded that the octagonal nutrient warnings performed best, compared to circular and triangular nutrient warnings [27], there is a dearth of evidence demonstrating which type of front-of-package label performs best in Colombia. There is also no information as to whether the impact of the front-of-package labels on food and drink selections varies by education among Colombians. This is important to know given concerns that front-of-package labeling systems may have less benefit for people with lower socio-economic status [28].

To inform this discussion, we originally published results from a randomized experiment testing front-of-package labels in Colombia in PLOS ONE in 2022 (doi: 10.1371/journal.pone.0263324) [29], and after publication, the study authors and a reader, independently noticed and then notified PLOS ONE about errors in the calculation of the nutritional information used in the mock front-of-package labels used in this study. We decided to retract our original article [*cite retraction once published*] and replicated the original study following an identical study protocol and identical sampling procedures, using corrected nutrition labels. This replication study included an additional quality control process in which all nutrition labels were independently created by two dietitians and then compared to ensure all steps of the nutrition label creation process were correct.

As in the original study, the objectives of this study were to identify the impact of nutrient warnings on participants' selection of "high in" products and ability to identify them, compared to GDA labels, Nutri-Score labels, and a no-label condition in Colombia. Specifically, the primary outcomes were 1) selection of the product high in sugar as the product the participant would rather buy, and 2) correct identification of the product higher in sugar. Secondary outcomes included perceived message effectiveness of the labels, likelihood of purchasing the product in the next week, ability to identify the less healthy product, ability to identify the product with excess of a nutrient of concern, and which label was perceived as most discouraging. We also investigated whether the primary outcomes varied by education level.

## Methods

Prior to launching data collection, we pre-registered the design, hypotheses, and analytic plan on ClinicalTrials.gov (#NCT05783726). Procedures and analyses in this study were identical to the previously published and retracted study, with two exceptions: 1) nutritional profiles and corresponding front-of-package labels were updated to correct values, and serving sizes for two products were updated to more closely reflect typical servings in Colombia; and 2) we

measured and reported one additional survey item (intentions to purchase breakfast cereals) that was not reported in the previous study. We report this study according to the CONSORT statement (**S1 Checklist**).

### Ethics statement

The online randomized study was approved by the institutional review board at the University of North Carolina at Chapel Hill (#20–0401) and designated as exempt from review at Universidad Nacional de Colombia. Prior to participating in the study, participants read the consent form and provided online written informed consent by proceeding onto the study.

### Study design and procedures

We selected labels to test in this study based on which front-of-package labeling systems were most relevant in the public discussion relating to the passing and implementation of Colombia's labeling law: nutrient warnings, which were proposed by advocacy coalitions and governmental groups, the GDA label (which was promoted by the food industry), and the Nutri-Score label (also promoted by the industry and used in Europe). We hypothesized that nutrient warnings would perform the best at helping consumers identify and reduce selection of foods high in nutrients of concern, as previous research has found nutrient warnings to perform best on such outcomes [30]. The labels tested are shown in **Fig 1**.

We selected the octagonal nutrient warnings because they performed best in our previous randomized experiment (compared to circular and triangular nutrient warnings and a control) which investigated front-of-package nutrient warnings [27]. The octagonal nutrient warnings were the label type that most participants selected as discouraging them from purchasing foods and sugary drinks high in nutrients of concern and were scored with the highest perceived message effectiveness (PME) [27]. The nutrient warning tested in this study was a black octagon that contained a statement about the product containing excess of a nutrient of concern (sugar, sodium, or saturated fat). For example, "EXCESO DE AZÚCARES" (Excess sugar). The octagon also contained "MINSALUD" indicating the message was authorized by the Colombian Ministry of Health and the text "EVITAR SU ALTO CONSUMO" (Avoid high consumption). To determine if a product would receive a nutrient warning, we used the final stage nutrient thresholds for sugar, sodium, and saturated fat from Chile's nutrient warning label law [31].

The GDA label included Spanish text above the GDA figure stating the product serving size. Below the serving size, a row of light blue blocks listed the Calories, total fat, saturated fat, sugar, and sodium per serving, as well as percentages indicating what percent of the GDA the serving contained. Underneath the light blue blocks, Spanish text explained the percentages were based on the guideline daily amounts for a 2,000-Calorie diet [32]. The GDA label was the only label condition that displayed the nutrition information of the product.

The Nutri-Score label system, which is currently used voluntarily in some European countries [33], is a color coded, letter rated (A-E) system. A dark green "A" indicates the healthiest nutritional value and a dark red "E" indicates the least healthy nutritional value. A product's letter rating is determined based on a point system. A higher point value indicates a less healthy product. The more calories, sugar, sodium, and saturated fat a product contains, the more points it receives. However, a product can also receive negative points for containing fiber, protein, and fruits and vegetables, which can decrease a product's total points [34].

Finally, we included a no-label condition. Our previous experiment of nutrient warnings in Colombia used a neutral barcode as a control in order to measure perceptions of and reactions to front-of-package labels [27]. However, in this study, we wanted to test actual policies that

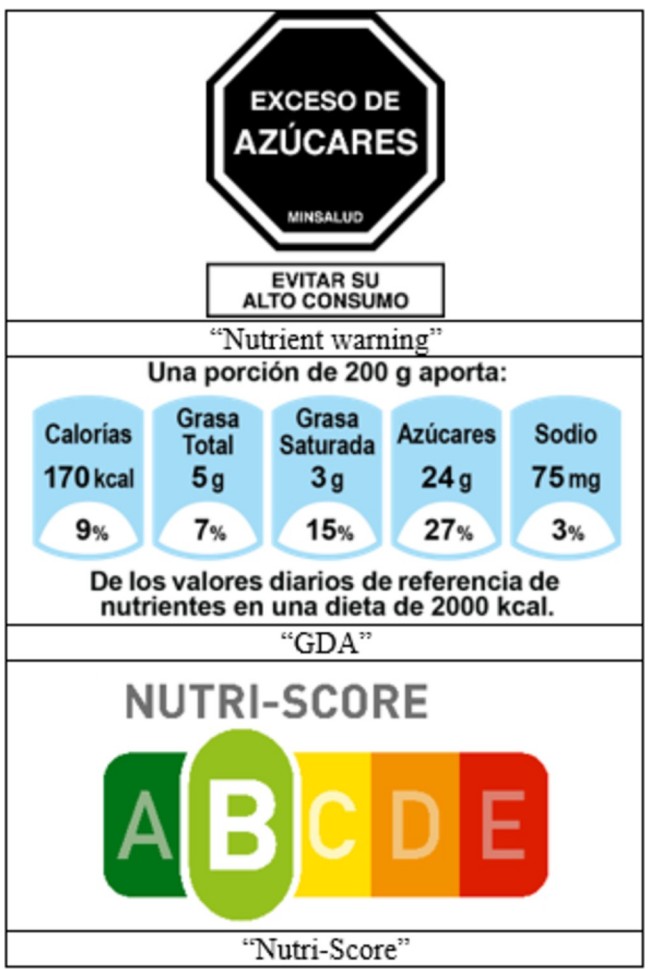

**Fig 1. Front-of-package labels used in experiment.** *Note.* Labels listed above represent the version of each label used on the yogurt (excess sugar).

could be implemented by the Colombian government. It was possible that the government could decide to not implement a front-of-package labelling system (status quo), so we also tested a no-label condition to measure the outcomes of maintaining the status quo compared to implementing the nutrient warnings. We used the Peruvian nutrient warning guidelines to design the size and placement of the label conditions [35].

## Product development and applied labels

We selected food and drink products from categories that make up some of the most commonly consumed ultra-processed foods in Colombia [17], and created nutrient profiles for the products modeled after real Colombian ultra-processed products that are high in nutrients of concern (sugar, sodium, and saturated fat). We used three products we had previously tested (fruit drink, oatmeal cookies, and sliced bread) [27], and a graphic designer developed three new products: a no-sugar-added fruit drink, breakfast cereal, and strawberry yogurt. The breakfast cereal contained excess amounts of both sugar and sodium, whereas the other products contained excess amounts of only one nutrient of concern. Therefore, the breakfast cereal

contained two nutrient warnings, while the other products only had one. All products contained fictional brand names to avoid the influence of consumer brand loyalty.

For each labelling system, the presence or absence of the label (nutrient warning) or content of the label (Nutri-Score, GDA) depended on the nutritional composition of the product. Table 1 provides each product's nutritional profile and the corresponding label applied. The back panels of the mock products were not visible; as such, there was no nutrition facts panel available for participants to review.

Prior to launching the replication study, our team implemented a quality control checklist to ensure accuracy of all stimuli and study procedures (S1 Table).

### Participants

From May 12, 2023 to June 22, 2023, we recruited adults in Colombia to participate in an experiment. We recruited participants through Offerwise, a market research company with over 300,000 panel participants in Colombia. Inclusion criteria included presently residing in Colombia and being between 18 and 65 years of age. We excluded panel members that participated in our previous study of front-of-package nutrient warnings in Colombia [27]. We set sample quotas for gender to reflect the Colombian population and for education level (half high school graduate or less, half college degree or higher) to ensure our sample was powered to detect differences in the primary outcomes by education level.

### Procedures

Participants completed an online survey programmed in Spanish using Qualtrics survey software. After providing informed consent, Qualtrics randomized participants using a simple 1:1:1:1 allocation ratio to one of the four front-of-package label conditions: nutrient warning, Nutri-Score, GDA, or a no-label condition. Participants then completed an online survey (measures described below). After finishing the survey, participants earned a pre-determined amount of points from Offerwise for completing the study. Participants are able to convert points into money once they accumulate a specified amount.

### Measures

Our study had two primary outcomes: 1) selection of the less healthy fruit drink as the fruit drink the participant would rather buy and 2) correctly identifying which fruit drink was higher in sugar. These outcomes were selected as primary outcomes because they are key steps on the pathway from labels to discouraging consumption of less healthy foods [30]. Secondary outcomes included the ability to identify the less healthy fruit drink, objective understanding, PME, intentions to purchase the products, and the most discouraging label. All measures were cognitively tested with Colombians of different education levels to make sure the measures were properly adapted to the Colombian context and accessible to all education levels. Measures can be found in S2 Table.

Participants first completed a fruit drink selection task, where they were asked a series of questions about two fruit drinks, one of which was healthier (contained 3 grams of naturally occurring sugar in the fruit and no added sugar) and one of which was less healthy (contained 18 grams of total sugars, including added sugar). We only included fruit drinks in the selection task due to survey space constraints. We used fruit drinks because of the increasing consumption of sugar-sweetened beverages in Colombia [15]. In the selection task, participants were asked to select which fruit drink was higher in sugar (*"Which of these products is higher in sugar?"*), which they would rather buy (*"Which of these products would you rather buy?"),* and which was most unhealthy (*"Which of these products is MOST unhealthy?"*). The fruit drinks

**Table 1. Product nutrition details and label applied to each product\*.**

| Mock Product | Nutrition profile | GDA Label (% of GDA) | Nutri-Score Label | Nutrient warning Label |
|---|---|---|---|---|
| No-added sugar fruit drink (450 mL) | Serving: 200mL<br>Calories: 15<br>Total fat: 0g<br>Saturated fat: 0g<br>Sugars: 3g<br>Sodium: 15mg<br>Proteins: 0g<br>Fiber: 1g<br>% Fruits, veg: 14% | Serving: 200mL<br>Calories: 1%<br>Total fat: 0%<br>Saturated fat: 0%<br>Sugars: 3%<br>Sodium: 1% | C | None |
| Fruit drink (450 mL) | Serving: 200mL<br>Calories: 75<br>Total fat: 0g<br>Saturated fat: 0g<br>Sugars: 18g<br>Sodium: 13mg<br>Proteins: 0g<br>Fiber: 0g<br>% Fruits, veg: 9% | Serving: 200 mL<br>Calories: 4%<br>Total fat: 0%<br>Saturated fat: 0%<br>Sugars:20%<br>Sodium: 1% | E | Excess sugar |
| Strawberry yogurt (200 g) | Serving: 200g<br>Calories: 170<br>Total fat: 5g<br>Saturated fat: 3g<br>Sugars: 24g<br>Sodium: 75mg<br>Proteins: 6g<br>Fiber: 2g<br>% Fruits, veg: 0% | Serving: 200g<br>Calories: 9%<br>Total fat: 7%<br>Saturated fat: 15%<br>Sugars: 27%<br>Sodium: 3% | B | Excess sugar |
| Oatmeal cookies (150 g) | Serving: 30g<br>Calories: 140<br>Total fat: 7g<br>Saturated fat: 3g<br>Sugars: 3g<br>Sodium: 40mg<br>Proteins: 2g<br>Fiber: 3g<br>% Fruits, veg: 6% | Serving: 30g<br>Calories: 7%<br>Total fat: 10%<br>Saturated fat: 15%<br>Sugars: 3%<br>Sodium: 2% | D | Excess saturated fat |
| Sliced bread (450 g) | Serving: 37g<br>Calories: 100<br>Total fat: 2g<br>Saturated fat: 1g<br>Sugars: 0g<br>Sodium:180mg<br>Proteins: 4g<br>Fiber: 1g<br>% Fruits, veg: 0% | Serving: 37g<br>Calories: 5%<br>Total fat: 3%<br>Saturated fat: 5%<br>Sugars: 0%<br>Sodium: 8% | C | Excess salt/sodium |
| Cereal (500 g) | Serving: 32g<br>Calories: 130<br>Total fat: 3g<br>Saturated fat: 0g<br>Sugars: 6g<br>Sodium: 135mg<br>Proteins: 2g<br>Fiber: 2g<br>% Fruits, veg: 0% | Serving: 32g<br>Calories: 7%<br>Total Fat: 4%<br>Saturated fat: 0%<br>Sugars: 7%<br>Sodium: 6% | C | Excess sugar; Excess salt/sodium |

\*Sugar values represent total sugar, including any naturally occurring and/or added sugars present in the product. Fruits, veg. refers to fruit, vegetables, legumes and nuts content.

were displayed with the labels corresponding to the participant's randomly assigned label condition. Both the order of the three questions and the order of the two fruit drinks (left or right) were randomized.

Next, participants completed single product assessment tasks. They viewed a prompt that read: "The next questions are about food products. You will look at a few different products and answer questions about each one. Please keep in mind that this study seeks to evaluate your survey responses and not the sale of the product." Then, they answered a series of questions about the yogurt, cookies, sliced bread, and breakfast cereals, shown according to the participant's randomly assigned label condition. In this section of the survey, participants answered all questions about one product at a time (with products displayed in random order). The breakfast cereal was always displayed last as the nutrient warning condition contained two labels.

The questions measured whether participants could correctly identify if the product contained excess of the nutrient of concern with the following item: "Do you think this product has excess [sugar, sodium, or saturated fat]?" (yes/no). We measured perceived message effectiveness (PME) of the labels using three items from the UNC PME scale [36, 37] which read: *"How much does the label. . ."* *"make you worried about the health consequences of consuming this product?"* (range from "not at all" (coded as 1) to "very much" (coded as 5)), *"make consuming this product seem unpleasant to you?"* (range from "not at all" (coded as 1) to "very much" (coded as 5)), and *"discourage you from wanting to consume this product?"* (range from "not at all" (coded as 1) to "very much" (coded as 5)). Because PME is specifically about labels, we did not measure PME for the no-label condition. The survey then assessed participants' likelihood of wanting to purchase the product in the next week if it were available, which read *"How likely is it for you to want to purchase this product next week, if it were available?"* (range from "not at all" (coded as 1) to "very much" (coded as 5)).

Finally, participants were randomly assigned by Qualtrics using a simple allocation ratio to see the yogurt, cookies, or sliced bread again (one product only). However, this time, the product did not include a front-of-package label. Instead, the three label types (nutrient warnings, GDA, and Nutri-Score) were displayed underneath the product and the participant was asked to select the most discouraging label. Participants were asked to select which label would most discourage them from wanting to consume the product (*"Which of these labels would discourage you <u>most</u> from wanting to consume this product?"*). At the end of the survey, participants answered standard demographic questions.

## Analyses

All analyses were conducted in STATA version 17.0. De-identified data used in this study and do files are available at https://osf.io/5vbzm. A two-sided critical alpha of 0.05 was used to assess statistical significance. Prior to the study, we used G.Power 3.1.9.4 to estimate that with a sample of 8,000 (2,000 per arm), alpha of 0.05, and 80% power, we would be able to detect a small effect of d = 0.09 for both primary outcomes (two-sided two-sample t-test for means).

We randomized 10,160 participants with a final analytical sample size of 8,004 (**Fig 2**). After randomization, 1,999 participants did not complete the study; we used per protocol analysis, so these participants were not included in our analytic sample. Additionally, 157 participants were excluded from analyses for any of the following reasons: missing all outcome data (n = 2), invalid age (n = 6), completing the study in less than two minutes (2), and repeat or overlapping responses (n = 147). In these cases, we dropped repeat responses (keeping each participant's first response) if they did not overlap and dropped all responses (including the first) if they did overlap. In the analytic sample, analyses were conducted on a complete-case

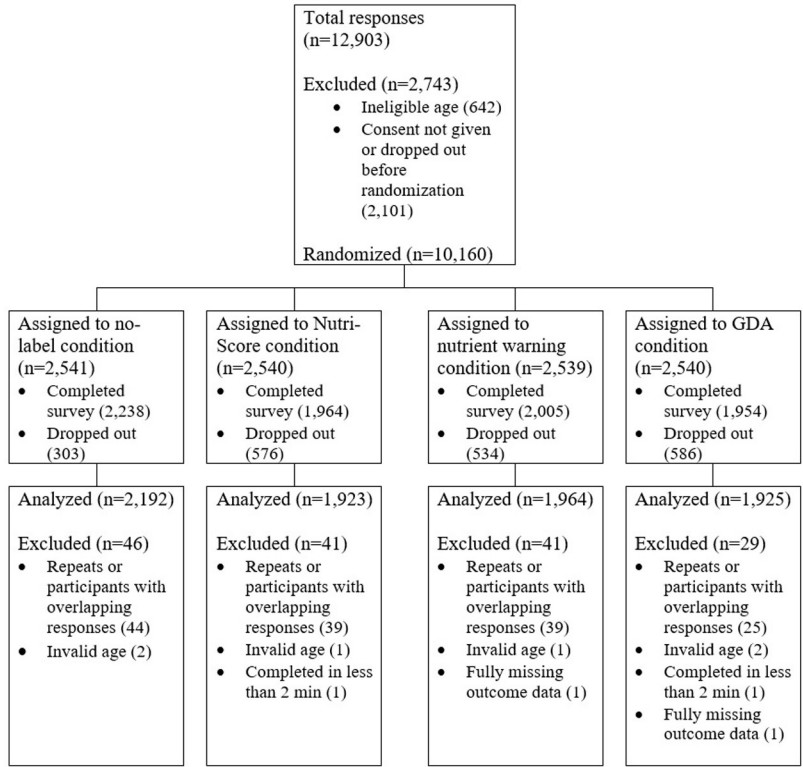

**Fig 2. CONSORT flow diagram.**

basis. In the single product assessment task where outcomes were analyzed at the participant-product level, we used all available data from any complete participant-product observation.

We calculated unadjusted means (and standard deviations) and percentages for the primary and secondary outcomes. For the secondary outcome of PME, we calculated it at the participant-product level as the average of the three items for each product type (Cronbach's alpha for each product type >.70) if all items were non-missing. PME was not calculated for participant-product observations with any PME item missing. We then assessed whether primary and secondary outcomes differed by condition compared to the nutrient warnings. We used linear regression for Likert-scale outcomes and PME (treated as continuous) and logistic regression for binary outcomes. For outcomes that were assessed using repeated measures for multiple product types, we used mixed models treating the intercept as random at the respondent level to adjust for repeated measures across multiple products. These models included the between-subjects factor (i.e., label type), the within-subjects factor (i.e., product type), and their interaction.

To evaluate the most discouraging label, we compared the proportions of participants who selected the Nutri-Score and the GDA labels as the one that most discouraged them from consuming products high in sugar, sodium, or saturated fat relative to the proportion who selected the nutrient warning label (pairwise proportions tests).

We applied Holm's sequentially rejective procedure [38] to all comparisons with the nutrient warnings across the primary and secondary outcomes to account for multiple comparisons.

Finally, to assess whether the effect of label type on the primary outcomes differed by education, we tested for an interaction of nutrient warnings with education level specified as low

(high school diploma or less) vs. high (college degree or higher) and used a Wald chunk test to determine the joint interaction. We conducted pairwise comparisons to predict percentages by label type and education level.

## Results

### Descriptive statistics

Participant characteristics are listed in Table 2.

### Fruit drink selection tasks

Selection of the less healthy fruit drink did not differ between the nutrient warnings and GDA labels; however, the nutrient warnings led to lower selection of the less healthy fruit drink compared to the Nutri-Score and no label conditions (Fig 3). Only 17% in the nutrient warning condition and 14% in the GDA label condition selected the less healthy fruit drink for purchase (adj p = 0.087), whereas 27% and 31% in the Nutri-Score and no-label conditions selected the less healthy fruit drink (both adj p<0.001 compared to the nutrient warning condition).

Relative to the nutrient warning condition, the GDA label was more effective at helping consumers identify which fruit drink was higher in sugar, although this difference was small (89% versus 92%, adj p<0.001). In contrast, the Nutri-Score condition and no-label conditions were less effective than the nutrient warnings. Only 67% in the Nutri-Score condition and 69% in the no-label condition correctly identified the fruit drink higher in sugar (both adj p<0.001 compared to the nutrient warnings).

When asked which fruit drink was less healthy, 89% in the nutrient warnings condition made the correct identification, compared to 72% in the Nutri-Score condition and 68% in the no-label condition (adj p<0.001 for both conditions compared to the nutrient warnings). There was no statistically significant difference between the nutrient warning and GDA label conditions.

### Interaction of label type and education, primary outcomes

There was no significant interaction between label condition and education level on selection of the fruit drink higher in sugar as the product the participant would rather buy (Wald test of joint significance for the condition-education level interactions: unadj p = 0.091) (S3 Table). There was also no significant interaction between label condition and education level on the likelihood of correctly identifying the fruit drink higher in sugar (Wald test of joint significance for the condition-education level interactions: unadj p = 0.267). Logistic regression estimates for the fruit drink selection task outcomes (with and without interactions with education level) can be found in S4 Table.

### Single product assessment of yogurt, bread, cookies, and cereal high in sugar, sodium, saturated fat, or sugar and sodium

In the single product assessment tasks, the nutrient warnings were more effective than the no-label, Nutri-Score, and GDA conditions in helping participants to correctly identify that the products contained excess of a nutrient of concern and more effective in decreasing the participants' likelihood of wanting to purchase the product if it were available (Table 3; adj p<0.001 for each condition compared to the nutrient warnings). While 79% of participants in the nutrient warnings condition correctly identified that the product contained excess of the nutrient of concern, only 33% in the no-label condition, 37% in the Nutri-Score condition, and 50% in

**Table 2. Socio-demographic characteristics of the sample (n = 8,004).**

| | No label (n = 2,192) | | Nutri-score (n = 1,923) | | GDA (n = 1,925) | | Nutrient warning (n = 1,964) | |
|---|---|---|---|---|---|---|---|---|
| | n | % | n | % | n | % | n | % |
| Age | | | | | | | | |
| 18–24 | 545 | 24.9 | 442 | 23.0 | 466 | 24.2 | 457 | 23.3 |
| 25–34 | 701 | 32.0 | 626 | 32.6 | 625 | 32.5 | 619 | 31.5 |
| 35–44 | 539 | 24.6 | 450 | 23.4 | 483 | 25.1 | 505 | 25.7 |
| 45–54 | 279 | 12.7 | 284 | 14.8 | 252 | 13.1 | 256 | 13.0 |
| 55–64 | 128 | 5.8 | 121 | 6.3 | 99 | 5.1 | 127 | 6.5 |
| Mean (SD) | 33.9 | (11.2) | 34.4 | (11.4) | 34.0 | (11.1) | 34.4 | (11.3) |
| Gender | | | | | | | | |
| Man | 1,084 | 49.5 | 904 | 47.0 | 957 | 49.7 | 958 | 48.8 |
| Woman | 1,101 | 50.2 | 1,012 | 52.6 | 959 | 49.8 | 1,004 | 51.1 |
| Other Gender Identity | 7 | 0.3 | 7 | 0.4 | 9 | 0.5 | 2 | 0.1 |
| Body-mass index (BMI, kg/m^2) | | | | | | | | |
| Underweight (10–18.49) | 146 | 6.7 | 126 | 6.6 | 130 | 6.8 | 122 | 6.2 |
| Normal weight (18.5–24.99) | 1,145 | 52.2 | 993 | 51.6 | 1,002 | 52.1 | 985 | 50.2 |
| Overweight (25–29.99) | 587 | 26.8 | 518 | 26.9 | 535 | 27.8 | 568 | 28.9 |
| Obese (30–79.99) | 252 | 11.5 | 223 | 11.6 | 193 | 10.0 | 239 | 12.2 |
| Missing or implausible | 62 | 2.8 | 63 | 3.3 | 65 | 3.4 | 50 | 2.5 |
| Mean (SD) | 25.2 | (7.6) | 25.1 | (7.2) | 24.8 | (6.7) | 25.3 | (7.2) |
| Education level | | | | | | | | |
| Secondary or lower | 1,017 | 46.4 | 820 | 42.6 | 829 | 43.1 | 870 | 44.3 |
| Tertiary | 1,175 | 53.6 | 1,103 | 57.4 | 1,096 | 56.9 | 1,094 | 55.7 |
| Region | | | | | | | | |
| Atlantica | 349 | 15.9 | 279 | 14.5 | 298 | 15.5 | 295 | 15.0 |
| Oriental | 328 | 15.0 | 319 | 16.6 | 295 | 15.3 | 313 | 15.9 |
| Central | 485 | 22.1 | 436 | 22.7 | 444 | 23.1 | 450 | 22.9 |
| Pacifica | 296 | 13.5 | 242 | 12.6 | 226 | 11.7 | 246 | 12.5 |
| Orinoquia | 34 | 1.6 | 23 | 1.2 | 25 | 1.3 | 23 | 1.2 |
| Bogota | 677 | 30.9 | 599 | 31.1 | 614 | 31.9 | 619 | 31.5 |
| Missing | 23 | 1.0 | 25 | 1.3 | 23 | 1.2 | 18 | 0.9 |
| Children in the household (0–18 years) | | | | | | | | |
| No | 717 | 32.7 | 634 | 33.0 | 613 | 31.8 | 666 | 33.9 |
| Yes | 1,463 | 66.7 | 1,273 | 66.2 | 1,293 | 67.2 | 1,286 | 65.5 |
| Missing | 12 | 0.5 | 16 | 0.8 | 19 | 1.0 | 12 | 0.6 |
| Ethnicity (all that apply) | | | | | | | | |
| Indigenous | 59 | 2.7 | 68 | 3.5 | 67 | 3.5 | 73 | 3.7 |
| African descendent | 149 | 6.8 | 112 | 5.8 | 146 | 7.6 | 133 | 6.8 |
| White | 615 | 28.1 | 545 | 28.3 | 576 | 29.9 | 562 | 28.6 |
| Mestizo | 828 | 37.8 | 711 | 37.0 | 663 | 34.4 | 722 | 36.8 |
| Other ethnic group | 153 | 7.0 | 105 | 5.5 | 128 | 6.6 | 125 | 6.4 |
| No ethnic group | 449 | 20.5 | 432 | 22.5 | 415 | 21.6 | 414 | 21.1 |
| Financial situation | | | | | | | | |
| Can pay the bills and buy additional things | 720 | 32.8 | 697 | 36.2 | 659 | 34.2 | 679 | 34.6 |
| Can pay the bills and buy what is needed | 943 | 43.0 | 800 | 41.6 | 793 | 41.2 | 801 | 40.8 |
| Can pay the bills but not buy what is needed | 361 | 16.5 | 279 | 14.5 | 331 | 17.2 | 346 | 17.6 |
| Can't pay the bills | 150 | 6.8 | 125 | 6.5 | 111 | 5.8 | 121 | 6.2 |
| Missing | 18 | 0.8 | 22 | 1.1 | 31 | 1.6 | 17 | 0.9 |

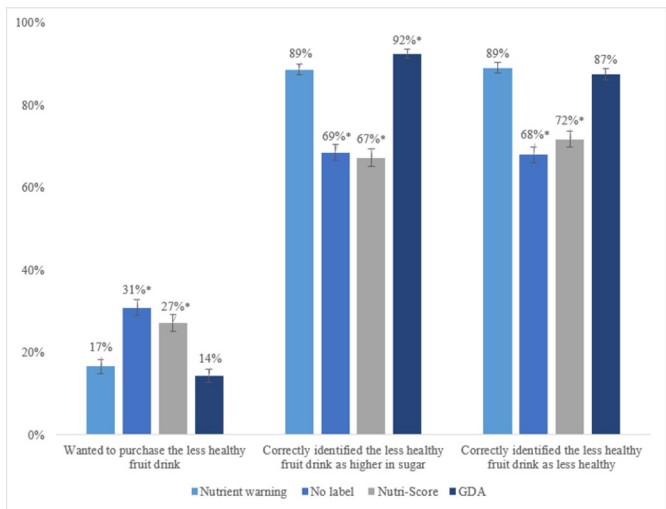

**Fig 3. Predicted percent, by label condition.** *Note.* * p<0.001 compared to nutrient warning (adjusted for 19 comparisons using Holm's sequentially rejective method). Data were analyzed by logistic regressions of the outcomes on indicator variables for the arm. Inference on the predicted percent is based on the delta method. Missing data at the participant level were as follows: 40 (0.5%) for 'Wanted to purchase the less healthy fruit drink' (11 in nutrient warning, 12 in no label, 9 in Nutri-score, and 8 in GDA), 16 (0.2%) for 'Correctly identified the less healthy fruit drink as higher in sugar' (4 in nutrient warning, 3 in no label, 8 in Nutri-score, and 1 in GDA), and 26 (0.3%) for 'Correctly identified the less healthy fruit drink as less healthy' (4 in nutrient warning, 15 in no label, 4 in Nutri-score, and 3 in GDA).

the GDA condition were able to do so (Table 3; adj p<0.001 for each condition compared to the nutrient warnings). The nutrient warnings also led to greater perceived message effectiveness (PME) compared to both the Nutri-Score and GDA (adj p<0.001 for each condition compared to the nutrient warnings; PME not assessed in the no-label condition). Unadjusted results on individual products can be found in **S5 Table**. Multilevel mixed-effects regression estimates for the single product assessment task outcomes can be found in **S6 Table**.

**Table 3. Predicted percent and predicted means of secondary outcomes by label type.**

| Condition | Correctly identified product as having excess nutrients | | | | Likelihood of purchasing the product in the next week if it were available | | | | Perceived message effectiveness (PME) | | | |
|---|---|---|---|---|---|---|---|---|---|---|---|---|
| | n | % | SE | p | n | Mean | SE | p | n | Mean | SE | p |
| Nutrient warning | 1964 | 79% | 0.58 | (ref) | 1959 | 2.77 | 0.02 | (ref) | 1962 | 3.75 | 0.02 | (ref) |
| No label | 2192 | 33% | 0.61 | < 0.001 | 2190 | 3.53 | 0.02 | < 0.001 | - | - | - | - |
| Nutri-score | 1923 | 37% | 0.71 | < 0.001 | 1919 | 3.41 | 0.02 | < 0.001 | 1920 | 2.89 | 0.02 | < 0.001 |
| GDA | 1925 | 50% | 0.65 | < 0.001 | 1923 | 3.29 | 0.02 | < 0.001 | 1924 | 3.18 | 0.02 | < 0.001 |

*Note.* n = number of participants with a valid response for at least one product. P-values adjusted for 19 comparisons with the nutrient warning using Holm's sequentially rejective method. Likelihood of purchasing and perceived message effectiveness ranged from 1 (lower) to 5 (higher). Data were analyzed by multilevel mixed-effects logistic ('Correctly identified product as having excess nutrients') and linear ('Likelihood of purchasing the product in the next week if it were available' and 'Perceived message effectiveness (PME)') regressions of the outcomes on indicator variables for the arm, for the product, and for arm-product interactions, with a random intercept at the participant level. Inference on the predicted percent and means is based on the delta method. Missing data at the participant-product level were as follows: 47 (0.1%) for 'Correctly identified product as having excess nutrients' (16 in nutrient warning, 5 in no label, 12 in Nutri-score, and 14 in GDA), 142 (0.4%) for 'Likelihood to purchase the product in the next week if it were available' (36 in nutrient warning, 23 in no label, 37 in Nutri-score, and 46 in GDA), and 116 (0.5%) for 'Perceived message effectiveness (PME)' (36 in nutrient warning, 38 in Nutri-score, and 42 in GDA).

## Other outcomes

Participants were most likely to select the nutrient warnings as the label that most discouraged them from wanting to consume a product high in sugar, saturated fat, or sodium (**Fig 4**). Seventy percent of participants selected the nutrient warnings as most discouraging compared to only 19% selecting the GDA label and 11% selecting the Nutri-Score label (adj p<0.001 for both conditions compared to the nutrient warnings).

## Discussion

This online experiment aimed to assess the impact of nutrient warnings on product selection and identification of less healthy products, compared to GDA, Nutri-Score, and no label conditions, among Colombian adults. We found that the nutrient warning performed better than the Nutri-Score and no label condition across outcomes. The nutrient warning performed similarly to or slightly worse than the GDA label on outcomes related to selection between two drinks, including selecting the less healthy drink, correctly identifying the high-sugar drink, and identifying which drink was less healthy. However, the nutrient warning label outperformed the GDA label as well as the Nutri-Score and no label conditions on outcomes assessed for multiple products (assessing one product at a time), including correctly identifying those products contained excess nutrients of concern, intentions to purchase products, and perceived message effectiveness. The nutrient warnings were also most frequently selected as the label that most discouraged consumption of nutrients of concern. The pattern of results illustrates the promise of nutrient warnings as a policy to help Colombian consumers identify and reduce consumption of products high in nutrients of concern.

While this study assessed only consumer reactions, not behaviors, the results are broadly consistent with the literature on nutrient warnings' impact on behavior, including results from a systematic review and meta-analysis of experiments [39], which found that nutrient warnings are effective at reducing purchases of sugar-sweetened beverages. The results are also consistent with evaluations of real-world nutrient warning label policies, which have found that such

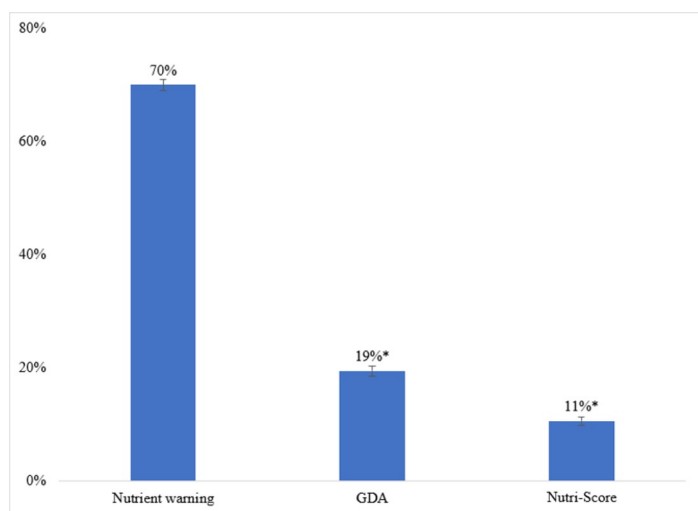

**Fig 4. Percent of participants selecting each label type as the most discouraging.** *Note.* * p<0.001 compared to nutrient warning (adjusted for 19 comparisons using Holm's sequentially rejective method). Data are estimated proportions and their logit-transformed 95% confidence intervals. Missing data at the participant level were as follows: 131 (1.6%) (22 in nutrient warning, 27 in no label, 39 in Nutri-score, and 43 in GDA).

policies, in conjunction with other policies, have led to reductions in purchases of foods and drinks high in nutrients of concern [27, 40].

Prior studies have found that GDA labels have performed worse than nutrient warnings [41, 42]. Indeed, our hypothesis was that GDA would perform worse than nutrient warnings in this study. However, the GDA and nutrient warnings performed similarly on outcomes for the drink product selection tasks, and the GDA was slightly better than nutrient warnings at helping consumers identify which of the two fruit drinks was high in sugar (92% vs. 89% of consumers, respectively). One possible reason for the relatively strong performance of the GDA could be due to the design of the selection task. In comparing only two products using the GDA in an online setting, consumers could see that sugar in one drink (20% daily value) was higher than sugar in the other drink (3% daily value). However, in a real life shopping situation, consumers' ability to use the GDA to identify "high-in" products could be limited, as people have little time to examine labels in detail, need to examine many products at once, and the numerical information provided in the GDA can be difficult to understand quickly [43].

On the other hand, when assessing one product at a time, for multiple products, we found that nutrient warnings performed better than the GDA label across outcomes, including identifying whether a product was high in nutrients of concern. One possibility is that consumers' ability to use the GDA could be limited if other products are not available for comparison; in other words, there are no other products to compare numerical values to. In contrast, nutrient warnings, which provide a binary signal about nutrient content, do not require other products to be available as a frame of reference for understanding the nutritional profile of a product. In short, the ability to use a given label type may be affected by the number of products being compared. Future research that investigates the performance of front-of-package labels in more realistic food shopping environments with a variable number of products would shed light on this question.

Our study found that nutrient warnings consistently performed better than Nutri-Score labels across all outcomes. The findings of this study are in contrast to some previous studies that have found more favorable outcomes for Nutri-Score [44, 45]. However, such studies have typically involved ranking tasks, while this study was concerned with identifying the less healthy products and the product that was "high in" the nutrient of concern. It is intuitive that Nutri-Score would perform worse than nutrient warnings on this outcome because the Nutri-Score provides a single summary measure or grade on the overall nutritional profile, rather than information about specific nutrients. We selected identification of "high in" products as primary outcomes, rather than ability to rank products, to align with the overarching public health goal of labeling laws in Colombia, which is to help consumers understand and reduce purchases of unhealthy foods.

This study found no evidence that nutrient warnings had differential effects by education level. In other words, it seems unlikely that Colombian consumers with high educational levels would benefit more from a nutrient warning policy than low-educated consumers. However, in the real world, differences in other factors like price or consumer preferences for certain foods may interact with the labeling law to create differences in response by socio-economic status. As nutrient warnings are implemented in Colombia, evaluation research should monitor whether the law differentially impacts consumer understanding and food purchases for high- vs. low-educated consumers.

## Strengths and limitations

Strengths of this study include between-subjects randomization allowing for causal inference, the use of standardized questions from previous studies, which have shown appropriate

psychometric characteristics [27, 36, 37], as well as the assessment of multiple products for several of our outcomes.

However, one important limitation is that this experiment was conducted during the process of implementation of nutrient warnings in Colombia. Participants may have already been exposed to circular warnings (implemented in a preliminary stage by the Colombian Ministry of Health) as well as octagonal warnings (similar to those used in this study and implemented under the new law) at the time of the study. It is possible that this prior exposure may have amplified or weakened the impact of the warning. Participants may have also been exposed to the GDA warnings, which are voluntarily implemented by the food industry and have been present in the Colombian food supply for many years. However, prior exposure is unlikely to have affected experimental findings due to randomization. More broadly, our results showing the effectiveness of nutrient warnings relative to other label types are consistent with experimental results in countries prior to implementation of nutrient warnings [43, 46–48] lending confidence to these findings. Secondly, there may be limitations to generalizability of study results to the general Colombian population due to our use of an online sample; we sought to mitigate this by including socioeconomically diverse participants from different Colombian regions.

## Conclusion

Colombian advocacy groups working on the labeling law in Colombia, as well as international scholars, have emphasized reducing excess consumption of unhealthy foods high in nutrients of concern as a key first step towards obesity prevention [49–51]. In this experiment, nutrient warnings consistently performed better than Nutri-Score and no label conditions at helping consumers to identify products high-in nutrients of concern and discourage their purchases. Nutrient warnings performed similar or slightly worse than the GDA label on outcomes when participants picked between two products and better than the GDA label on outcomes when participants saw one product at a time. Compared to all three label types, nutrient warnings were perceived as most effective and were most frequently selected as the label that most discouraged participants from wanting to consume a less healthy product. The overall results suggest that nutrient warnings are a promising policy for helping Colombian consumers identify and reduce the consumption of unhealthy foods. Given that nutrient warnings are now being implemented in Colombia, evaluation research is needed to understand the impact of nutrient labels on actual "high in" food and beverage purchases, as well as to evaluate the impact across populations with differing socioeconomic status.

## Supporting information

**S1 Checklist. CONSORT 2010 checklist of information to include when reporting a randomised trial\*.**
(DOC)

**S1 Table. Quality control checklist.**
(DOCX)

**S2 Table. Survey measures and responses.**
(DOCX)

**S3 Table. Predicted probabilities between high and low education by outcome and condition.** \*P-values (except for the joint significance of the interaction terms) are for the difference in the contrast with nutrient warning (the reference) by education level (e.g., for no label, the difference between the contrast for secondary or lower, 30.4–16.2, and the contrast for tertiary,

31.3–17). Thus, the effects of no label, Nutri-score, and GDA relative to nutrient warning did not significantly differ by education level. Data analyzed by logistic regressions of the outcomes on indicator variables for the arm, education level, and their interactions. Inference on the contrasts is based on the delta method. Missing data were as follows: 40 (0.5%) for 'Wanted to purchase the less healthy drink' (11 in nutrient warning, 12 in no label, 9 in Nutri-score, and 8 in GDA) and 16 (0.2%) for 'Correctly identified the less healthy fruit drink as higher in sugar' (4 in nutrient warning, 3 in No label, 8 in Nutri-score, and 1 in GDA).
(DOCX)

**S4 Table. Logistic regression estimates for the fruit drink selection task outcomes (with and without interactions with education level for the primary outcomes).** Standard errors in parentheses. Data analyzed at the participant level.
(DOCX)

**S5 Table. Unadjusted means and proportions by label type for single product assessment task.**
(DOCX)

**S6 Table. Multilevel mixed-effects regression estimates for the single product assessment task outcomes.** Standard errors in parentheses. PME not measured in No label. No residual variance for 'Correctly identified product as having excess nutrients' (multilevel mixed-effects logistic regression). Data analyzed at the participant-product level.
(DOCX)

## Acknowledgments

The authors would like to thank the UNC Global Food Research Program and Javeriana team. In particular, we thank Dr. Barry Popkin for his insight on the global food policy landscape, Dr. Yazmin Cadena for advising and providing insight into the cognitive interview process, Emily Busey for her assistance developing the images used in this study, Carmen E. Prestemon for programming the survey and assisting with formatting the manuscript, and Cindy P. Evans for assisting in the development of nutritional profiles for all products.

## Author Contributions

**Conceptualization:** Mercedes Mora-Plazas, Luis Fernando Gomez, Marissa G. Hall, Lindsey Smith Taillie.

**Formal analysis:** Isabella Carolyn Aida Higgins, Maxime Bercholz.

**Funding acquisition:** Lindsey Smith Taillie.

**Investigation:** Mercedes Mora-Plazas, Luis Fernando Gomez, Marissa G. Hall, Maria Fernanda Parra, Nandita Murukutla, Lindsey Smith Taillie.

**Methodology:** Mercedes Mora-Plazas, Isabella Carolyn Aida Higgins, Luis Fernando Gomez, Marissa G. Hall, Maria Fernanda Parra, Nandita Murukutla, Lindsey Smith Taillie.

**Project administration:** Isabella Carolyn Aida Higgins.

**Writing – original draft:** Mercedes Mora-Plazas, Isabella Carolyn Aida Higgins, Luis Fernando Gomez, Maria Fernanda Parra, Lindsey Smith Taillie.

**Writing – review & editing:** Mercedes Mora-Plazas, Isabella Carolyn Aida Higgins, Luis Fernando Gomez, Marissa G. Hall, Maria Fernanda Parra, Maxime Bercholz, Nandita Murukutla, Lindsey Smith Taillie.

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
