## [Decision Letter · Decision Letter 0]

16 Feb 2024

PONE-D-23-37686Impact of nutrient warning labels on Colombian consumers’ selection and identification of food and drinks high in sugar, sodium, and saturated fat: A randomized controlled trialPLOS ONE

Dear Dr. Taillie,

Thank you for submitting your manuscript to PLOS ONE. After careful consideration, we feel that it has merit but does not fully meet PLOS ONE’s publication criteria as it currently stands. Therefore, we invite you to submit a revised version of the manuscript that addresses the points raised during the review process. Please submit your revised manuscript by Apr 01 2024 11:59PM. If you will need more time than this to complete your revisions, please reply to this message or contact the journal office at plosone@plos.org. Please include the following items when submitting your revised manuscript:A rebuttal letter that responds to each point raised by the academic editor and reviewer(s). You should upload this letter as a separate file labeled 'Response to Reviewers'.A marked-up copy of your manuscript that highlights changes made to the original version. You should upload this as a separate file labeled 'Revised Manuscript with Track Changes'.An unmarked version of your revised paper without tracked changes. You should upload this as a separate file labeled 'Manuscript'.If applicable, we recommend that you deposit your laboratory protocols in protocols.io to enhance the reproducibility of your results. Protocols.io assigns your protocol its own identifier (DOI) so that it can be cited independently in the future. For instructions see: https://journals.plos.org/plosone/s/submission-guidelines#loc-laboratory-protocols. Additionally, PLOS ONE offers an option for publishing peer-reviewed Lab Protocol articles, which describe protocols hosted on protocols.io. Read more information on sharing protocols at https://plos.org/protocols?utm_medium=editorial-email&utm_source=authorletters&utm_campaign=protocols.

We look forward to receiving your revised manuscript.

Kind regards,

Rosely Sichieri

Academic Editor

PLOS ONE

Journal Requirements:

Additional Editor Comments:

I have only two comments:

1- Better explain the limitations due to use of an online sample. Groups are well balanced but information of how many of the 300,000 panel participants in Colombia were invited should be provided. Also, after randomization about 20% was lost (did not complete the questionaries). Exclusions only apply for those before randomization

2- I am unaware of using repeated measures for comparisons for different products as follows “ For outcomes that were assessed using repeated measures for multiple product types, we used mixed models treating the intercept as ……” These models included the between-subjects factor (i.e., label type), the within-subjects factor (i.e., product type), and their interaction. (table 3).

Supplementary table 4 is more informative than the table 3.

Correctly identified product as having excess nutrients. Values are almost the same: 79 32 37 50 vs 78.7, 32.5, 36.9 and 50.

I could not understand the meaning of the interaction in this analysis.

Reviewers' comments:

Reviewer's Responses to Questions

**Comments to the Author**

1. Is the manuscript technically sound, and do the data support the conclusions?

Reviewer #1: Partly

2. Has the statistical analysis been performed appropriately and rigorously? 

Reviewer #1: No

3. Have the authors made all data underlying the findings in their manuscript fully available?

Reviewer #1: Yes

4. Is the manuscript presented in an intelligible fashion and written in standard English?

Reviewer #1: Yes

5. Review Comments to the Author

Reviewer #1: Comments

Lines 129-132, the primary and secondary objectives are to be clearly outlined based on primary and secondary outcomes.

Line 305, the outcomes are to be clearly stated. Whether separate calculations for each primary outcome and between each arm were carried out and whether one with the largest sample size was chosen is to be clearly stated. More information is to be provided in this section.

A write-up on missing data is to be provided in the method section.

Per protocol analysis is to be stated.

Line 291, the type/name of random method is to be stated.

For Table 3 SE (pp), pp is to be omitted.

Linear regression, logistic regression, and mixed models which were mentioned in the analyses section in the methodology are to be presented with detailed output and denoted in the tables/figures footnotes in the results section.

S4 Table, n to be inserted apart from %.

S3 Table, n to be provided. The nutrient warning as a reference is to be indicated in the table footnote.

References did not conform to the journal format.

6. PLOS authors have the option to publish the peer review history of their article (what does this mean?). If published, this will include your full peer review and any attached files.

Reviewer #1: No

---

## [Author Response · Author response to Decision Letter 0]

1 Apr 2024

Dear editors and reviewers,

Thank you for your review and comments. We have included specific responses to your comments in the attached file titled "PLOS One Response to Reviewers_03.29.24.docx". We believe that these revisions have resulted in a stronger presentation of our work. We note a few major changes including:

(1) We have included further detail on how we reached the final analytic sample of 8,004.

(2) We have more clearly stated the primary and secondary outcomes of interest in the background section. 

(3) We have added additional information about missing data. 

We look forward to your further review.

Best,

Lindsey Smith Taillie on behalf of co-authors

---

## [Editor Report · Decision Letter 1]

26 Apr 2024

Impact of nutrient warning labels on Colombian consumers’ selection and identification of food and drinks high in sugar, sodium, and saturated fat: A randomized controlled trial

PONE-D-23-37686R1

Dear Dr. Taillie,

We’re pleased to inform you that your manuscript has been judged scientifically suitable for publication and will be formally accepted for publication once it meets all outstanding technical requirements.

Kind regards,

Rosely Sichieri

Academic Editor

PLOS ONE
---

## [Editor Report · Acceptance letter]

16 May 2024

PONE-D-23-37686R1 

PLOS ONE

Dear Dr. Taillie, 

I'm pleased to inform you that your manuscript has been deemed suitable for publication in PLOS ONE. Congratulations! Your manuscript is now being handed over to our production team.

Kind regards, 

on behalf of

Dr. Rosely Sichieri 

Academic Editor

PLOS ONE